# Climate Factors Affect Above–Belowground Biomass Allocation in Broad-Leaved and Coniferous Forests by Regulating Soil Nutrients

**DOI:** 10.3390/plants12233926

**Published:** 2023-11-22

**Authors:** Xing Zhang, Yongzhi Zhou, Yuhui Ji, Mengyao Yu, Xinyi Li, Jie Duan, Yun Wang, Jie Gao, Xiali Guo

**Affiliations:** 1Guangxi Key Laboratory of Forest Ecology and Conservation, College of Forestry, Guangxi University, Nanning 530004, China; zxyybh@163.com (X.Z.); heyzhouyongzhi@gmail.com (Y.Z.); 2College of Life Sciences, Xinjiang Normal University, Urumqi 830054, China; jyh1757838696@163.com (Y.J.); yao02292023@163.com (M.Y.); lxy0906xx@163.com (X.L.); duan0729lll@163.com (J.D.); www030517@outlook.com (Y.W.); 3Key Laboratory of Earth Surface Processes of Ministry of Education, College of Urban and Environmental Sciences, Peking University, Beijing 100871, China

**Keywords:** forest type, biomass allocation, temperature, precipitation, soil nutrient

## Abstract

The allocation of plant biomass above and below ground reflects their strategic resource utilization, crucial for understanding terrestrial carbon flux dynamics. In our comprehensive study, we analyzed biomass distribution patterns in 580 broadleaved and 345 coniferous forests across China from 2005 to 2020, aiming to discern spatial patterns and key drivers of belowground biomass proportion (BGBP) in these ecosystems. Our research revealed a consistent trend: BGBP decreases from northwest to southeast in both forest types. Importantly, coniferous forests exhibited significantly higher BGBP compared to broadleaved forests (*p* < 0.001). While precipitation and soil nutrients primarily influenced biomass allocation in broadleaved forests, temperature and soil composition played a pivotal role in coniferous forests. Surprisingly, leaf traits had a negligible impact on BGBP (*p* > 0.05). Climatic factors, such as temperature and rainfall, influenced biomass partitioning in both strata by altering soil nutrients, particularly soil pH. These findings provide valuable insights into understanding carbon sequestration dynamics in forest ecosystems and improving predictions of the future trajectory of this critical carbon cycle component.

## 1. Introduction

Forests, Earth’s largest terrestrial ecosystems, occupy approximately one-third of the planet’s land surface [1]. These expansive carbon sinks play a critical role in mitigating climate change by photosynthetically absorbing atmospheric carbon dioxide and storing it as biomass in both flora and soil. Forest biomass, which represents the most significant terrestrial carbon reservoir, is essentially the total mass of living flora per unit area, serving as a vital indicator of ecosystem productivity and carbon sequestration potential [2]. Comprising both aboveground (AGB) and belowground biomass (BGB), often measured as dry weight per unit area, biomass allocation reveals insights into plants’ strategic resource management strategies [3]. This understanding not only sheds light on forest carbon dynamics and plant adaptability [4], but also enhances predictions of terrestrial carbon retention, which is pivotal for global carbon cycling [5,6].

Biomass allocation within plants, which serves as an indicator of their environmental adaptability, is commonly assessed through the below-to-aboveground biomass proportion [2,7]. Despite the surge in forest biomass research, encompassing diverse forest categories and climatic regions [2], a significant research gap remains for pure coniferous and broadleaved forests (forests composed solely of coniferous or broadleaf species), with existing studies primarily focused on natural, planted, or mixed-species stands. Different forest types adopt distinct biomass allocation strategies, with coniferous forests maintaining the highest BGBP, surpassing their evergreen counterparts, while both evergreen and deciduous broadleaved forests exhibit lower ratios. Geographic variations also exist, as coniferous and deciduous broadleaved forests in China’s northeast and north tend to have higher BGBP, in contrast to lower ratios in southern forests [8]. Luo et al. [9] highlighted a significant BGBP contrast between coniferous and broadleaved forests, underscoring the pronounced differences in their biomass allocation strategies [10].

Amidst escalating global temperatures and intensified precipitation fluctuations, vegetation photosynthesis and resource availability, both above and below ground, are significantly influenced, leading to a rearrangement of vegetative biomass distribution. This recalibration has a substantial impact on terrestrial carbon cycling and its associated climate feedback mechanisms [3]. Biomass allocation among roots, stems, and leaves represents a complex balance between above-surface resources, such as light and CO_2_, and below-surface resources, like water and nutrients, which is critical for vegetation’s ability to adapt to climate changes [3]. Climate variables, especially mean annual temperature (MAT) and precipitation (MAP), play a crucial role in shaping plant BGBP [7,10]. Luo et al.’s [9] study revealed a negative correlation between MAT and BGBP, as well as a U-shaped relationship between MAP and BGBP in various forests, suggesting that arid conditions favor root growth, while heavy rainfall reduces nutrient availability through leaching and dilution. Globally, colder regions tend to allocate more nutrients to roots to compensate for reduced soil vitality [11]. Both angiosperm and gymnosperm forests in these cooler zones exhibit a preference for root-focused biomass allocation [12]. Alarmingly, even a modest 2.5 °C global temperature increase could shift biomass allocation towards roots, with precipitation significantly influencing vegetation’s response to temperature changes [6]. In drier landscapes, warming promotes root-centric biomass allocation, whereas in humid regions, the effect is less pronounced [3,6]. Precipitation changes also yield intricate consequences: increased rainfall generally reduces belowground biomass, while decreased rainfall enhances it [13,14]. Interestingly, root biomass appears to be less influenced by rainfall compared to its aboveground counterpart [10], and the partitioning of biomass between roots and leaves does not consistently mirror variations in drought intensity [12].

Plant biomass allocation, encompassing both above and belowground components, intricately interacts with crucial soil factors, including nitrogen, phosphorus, and soil pH. These elements are essential for vegetative growth, with soil pH regulating nutrient accessibility [15]. The optimal allocation theory posits that resource limitations, particularly belowground resources, drive plants to prioritize the expansion of root biomass, a strategy pronounced in cold and arid environments [3,12]. While some discussions suggest that environmental changes have a mild impact on plant biomass distribution, advocating for the allometric growth theory for deeper insights [16], the majority of evidence supports the optimal allocation paradigm. Clearly, plants in nitrogen-deficient soils increase their root mass fraction, in contrast to those in nitrogen-rich environments [17]; nitrogen supplementation typically reduces root biomass [13,14]. Additionally, coniferous forests exhibit distinct differences in biomass allocation compared to broadleaved forests. Broadleaved species, with more interconnected leaf trait networks, may adapt more adeptly to environmental fluctuations [18]. Coniferous counterparts, resilient to environmental challenges, allocate more nutrients to photosynthetic foliage, allocating less to trunks and branches compared to broadleaved species [11]. In alpine coniferous ecosystems at extreme latitudes or altitudes, nitrogen scarcity becomes prominent, constrained by cold temperatures, seasonal soil freezing, nutrient deficiency, and shortened growing seasons [19]. In contrast, broadleaved forests frequently contend with phosphorus as the primary limiting factor [15].

In modern research, plant functional traits have become essential for predicting and explaining crucial ecological dynamics, particularly when assessing plant responses to climate variability and their implications for forest biomass distribution [20,21]. These traits, which accurately signal vegetative adaptations to environmental changes, have significant implications for ecosystem functions, especially when considered within the broader environmental context [22].

The interplay between functional traits and environmental parameters critically informs plant growth, resource acquisition efficiency, and utilization, significantly influencing biomass production and distribution in forest ecosystems [23]. A multitude of studies have elucidated how climatic factors and soil characteristics intricately regulate plant functional traits. For instance, global analyses indicate that regions experiencing elevated peak annual temperatures and intense solar radiation often develop foliage with increased thickness and leaf dry matter content (LDMC) [24]. Areas characterized by high temperatures, aridity, and strong sunlight exhibit higher leaf nitrogen (LN) levels but lower specific leaf area (SLA) and reduced photosynthetic capacity [25]. Furthermore, rising temperatures and extended growing seasons, particularly in tropical regions, are associated with a decrease in leaf nitrogen and phosphorus content, along with an increasing nitrogen-to-phosphorus ratio [26]. These dynamic interactions between inherent traits (such as functional traits) and external factors (like climatic conditions and soil fertility) play a crucial role in shaping biomass allocation in plant structures, including both aboveground and belowground components. Plants with lower SLA and higher LDMC, adapted for resilience in nutrient-poor, arid environments, often prioritize biomass allocation to root systems, enhancing resource capture and environmental resilience. In contrast, plants in regions with adequate water, temperature, and resource availability typically exhibit higher SLA and lower LDMC, directing biomass preferentially to photosynthetic structures, thereby promoting aboveground growth. This strategic balance among functional traits represents the result of extensive environmental adaptation and underscores the nuanced resource acquisition strategies of different plant species. Indeed, functional traits provide a valuable framework for understanding patterns in plant biomass allocation. Notably, species characterized by extensive SLA and high LN levels, despite having lower LDMC, tend to have greater aboveground biomass production, even in nitrogen-poor soil niches, highlighting the complex interplay between trait expressions and resource access [27]. Moreover, certain leaf characteristics appear to be adapted for enhanced phosphorus utilization in nutrient-deficient environments [28]. It is essential to acknowledge, however, that the effectiveness of functional traits in elucidating plant biomass distribution varies considerably across diverse geographical regions and plant categories. Thus, uncovering the specifics of how these traits illuminate biomass allocation in coniferous and broadleaved forests across extensive spatial scales remains an active area of scholarly investigation.

The distribution of plant biomass, both aboveground and belowground, is influenced by a complex interplay of factors. Exploring the drivers of biomass distribution in diverse forest ecosystems, including both coniferous and broadleaved forests, is crucial for enhancing our understanding of forest resource utilization strategies in the face of global environmental changes. To investigate the nuances of biomass allocation strategies and their underlying determinants in broadleaved and coniferous forests, we compiled a comprehensive dataset encompassing climatic variables, soil nutrient indices, and plant functional traits from 925 forest sites across China. Our research objectives aimed to address several hypotheses: (1) Substantial differences exist in biomass allocation patterns between broadleaved and coniferous forests. (2) Precipitation plays a pivotal role in shaping biomass distribution in both forest types. (3) Climatic factors predominantly influence both aboveground and belowground biomass allocation by regulating soil nitrogen and phosphorus levels, in conjunction with soil pH. Through this scholarly perspective, we aim to shed light on the intricate mechanisms governing biomass allocation in diverse forest ecosystems, thus enhancing our understanding of forest adaptability and responsiveness within the evolving global landscape.

## 2. Results

### 2.1. Geographical Patterns of BGBP

The spatial dispersion of BGBP across Chinese broadleaved and coniferous realms revealed a pronounced gradient, with apex values in the northwest descending toward the southeast (Figure 1A). Moreover, a stark contrast (*p* < 0.001) in BGBP between the two forest types was manifest, as coniferous expanses significantly eclipsed their deciduous counterparts in biomass proliferation (Figure 1B).

### 2.2. BGBP and Functional Traits

The first two principal components (PC1 and PC2) collectively accounted for 56.39% of the variation in functional traits. PC1 predominantly reflected leaf N/P and SLA, while PC2 was primarily influenced by LN and LP (Figure 2). In broadleaved forests, BGBP exhibited a noteworthy positive correlation with trait PC2 (*p* < 0.001). Conversely, in coniferous forests, this correlation with trait PC2 was relatively weak. Furthermore, no significant correlation was observed between BGBP and trait PC1 in both broadleaved and coniferous forests (Figure 3).

### 2.3. Relationships between Climate Factors, Soil Factors, and BGBP

With the increase in MAT and MAP, BGBP decreased significantly in both broadleaved and coniferous forests, showing a similar trend of change (*p* < 0.001) (Figure 4). The BGBP of both broadleaved and coniferous forests was positively correlated with soil pH (Figure 5). In broadleaved forests, the BGBP decreased significantly with increasing soil P but had no significant correlation with soil N (*p* < 0.001) (Figure 5A,B). The BGBP in coniferous forests was not significantly correlated with both soil N and soil P (*p* > 0.05) (Figure 5A,B).

For both broadleaved and coniferous forests, MAT and MAP exerted a negative influence on BGBP. However, it is noteworthy that in the case of broadleaved forests, the negative coefficient effect on BGBP was more pronounced than that observed in coniferous forests (Figure 6). Furthermore, soil pH exhibited a positive effect on BGBP in both broadleaved and coniferous forests, with the positive coefficient effect on BGBP being greater in broadleaved forests than in coniferous ones. In the context of broadleaved forests, soil phosphorus (P) content had a negative impact on BGBP, while this effect was not statistically significant in coniferous forests (Figure 6).

The heatmap analysis further revealed that BGBP in broadleaved and coniferous forests was significantly associated with climate and soil nutrient factors (Figure 7). Among these related factors, the BGBP of broadleaved forests had the strongest correlation with soil pH and MAP, while that of coniferous forests had the strongest correlation with soil pH and MAT.

The path analysis showed that climate and soil nutrient factors were the main driving factors affecting BGBP in broadleaved and coniferous forests (Figure 8). Temperature and precipitation could directly affect BGBP but indirectly affected BGBP mainly by regulating soil nutrient factors (especially soil pH). In broadleaved forests, BGBP was mainly affected by precipitation and soil nutrient factors. However, in coniferous forests, BGBP was mainly affected by temperature and soil nutrient factors. Functional traits had no significant effect on BGBP in broadleaved forests (*p* > 0.05). SEM explained 95% of the BGBP variation in broadleaved and coniferous forests.

## 3. Discussion

The allocation of biomass, both above and below ground, within plant populations reflects ecosystem productivity, environmental adaptability, and the delicate balance in resource utilization between terrestrial and subterranean domains [2,3]. In the context of a changing global climate, fluctuations in resource availability trigger a realignment of biomass distribution in plant communities [3]. In regions characterized by cold temperatures, aridity, and limited belowground resources, vegetation tends to strengthen its belowground structure, directing a higher proportion of biomass into the root system. This strategic adjustment enhances resource acquisition and resilience, supporting continued vegetative growth [3,11,12]. Conversely, areas with abundant moisture and temperate climates often witness plants allocating a larger portion of resources to aboveground biomass, promoting rapid growth. As a reflection of these principles, China’s geographic landscape of belowground biomass proportion (BGBP) demonstrates a decreasing trend from northwest to southeast, with central and northwestern regions—characterized by their arid climate and cooler temperatures—exhibiting higher BGBP, in contrast to the lower values observed in the warm, moisture-rich southern areas (Figure 1A). This distribution aligns with the established pattern of decreasing BGBP associated with increases in mean annual temperature (MAT) and mean annual precipitation (MAP) in our study, consistent with findings from extensive scholarly research [6,8,9,14,29].

These distinctions in tree characteristics manifest as profound differences in foliar architecture, morphological dimensions, dendrological form, and adaptive strategies tailored to their respective ecological niches, as elucidated by Li et al. [18]. Notably, a significant contrast in belowground biomass proportion (BGBP) is evident between these two forest categories, with coniferous forests surpassing their broadleaved counterparts in terms of biomass productivity. This trend is visually depicted in Figure 1B. This finding underscores the substantial divergence in biomass allocation strategies between the two forest types, potentially rooted in their intrinsic physiological growth traits and adaptability to different environmental conditions. Coniferous tree species, for instance, exhibit greater resilience in arid and cold climates. To enhance their survival in such challenging conditions, these trees invest heavily in root development, leading to a greater allocation of biomass belowground. This evolutionary adaptation optimizes moisture and nutrient absorption, reinforcing their ability to thrive in harsh, resource-scarce landscapes. In contrast, broadleaved forests tend to prioritize biomass allocation to aboveground structures, a strategy largely attributed to anatomical differences that distinguish broadleaved species from their coniferous counterparts.

Leaf functional traits, which play a fundamental role in governing key arboreal functions such as carbon assimilation, hydraulic circulation, and nutrient cycling, hold a heightened significance within the ecological context. Interestingly, broadleaved forests exhibit a more intricate network of interdependencies among leaf traits compared to coniferous forests. This suggests a greater adaptational capability in response to environmental changes, a hypothesis that aligns with our empirical observations [18]. Our analysis reveals an intriguing nuance: the varying contributory emphasis of trait PC1, primarily influenced by specific leaf area (SLA) and leaf N/P ratios, within the trait-based PCA analysis for broadleaved and coniferous biomes. This disparity appears to correspond with the marked morphological and textural differences observed in the foliage of the two forest types. Furthermore, trait PC2 exhibits a positive correlation with belowground biomass proportion (BGBP) in broadleaved ecosystems, as depicted in Figure 3, affirming its positive impact on biomass accumulation in these environments. However, the influence of trait PC1 and PC2 on BGBP within coniferous forests remains less clear, lacking statistically significant associations in either direction, as corroborated by Figure 6, with their interrelationship similarly lacking significant correlation. In summary, these findings highlight the divergent spectra of leaf traits at play in broadleaved and coniferous forests. Broadleaved ecosystems, compared to their coniferous counterparts, appear to be more responsive to environmental fluctuations, exhibiting flexibility that potentially underpins more robust ecological resilience.

Plants utilize their aerial structures to capture sunlight and carbon dioxide, enabling the synthesis of essential organic molecules. Simultaneously, their subterranean components absorb vital water and minerals, playing a critical role in growth and overall vitality [30]. The availability of nitrogen and phosphorus, key elements in these processes, depends on a complex interplay of environmental factors. Extreme drought and cold conditions can reduce moisture levels and hinder nutrient release and absorption, particularly affecting phosphorus uptake. Interestingly, prolonged drought conditions can lead to an increase in accessible soil phosphorus, paradoxically resulting in reduced plant and mycological β-diversity [31].

Additionally, soil pH plays a crucial role in regulating nutrient availability. Phosphorus is more abundant in neutral soils but becomes less available in acidic or alkaline environments due to interactions with elements like iron, aluminum, or calcium. Plants, with their adaptive mechanisms, adjust their nitrogen and phosphorus acquisition strategies by controlling their development and forming symbiotic relationships with microbes. This ability to sense and respond to nutrient availability shapes their symbiotic interactions with the environment, ultimately influencing nutrient uptake and growth patterns [30].

Historical research has indicated a dichotomy where phosphorus limits the growth of broadleaved trees, while nitrogen constrains coniferous counterparts [15]. In broadleaved ecosystems, increased phosphorus availability enhances metabolic efficiency, leading to greater aboveground biomass and reduced belowground biomass. As a result, a clear relationship emerges between soil P content and BGBP in these areas, with higher soil P content inversely affecting BGBP. Additionally, soil pH, a critical factor in nutrient dynamics, exerts significant influence over plant nutrient uptake, primary productivity, and biomass allocation [5]. Our findings emphasize the pivotal role of soil pH in shaping BGBP in both types of forests, highlighting its substantial connection with BGBP regardless of forest type.

The distribution of plant biomass is the result of a complex interplay involving both intrinsic and extrinsic factors. While climate and soil conditions have traditionally been recognized for their significant role, the influence of functional traits is now being increasingly acknowledged. These traits, which exhibit substantial intra-species diversity in response to environmental variations, are subject to global variations driven by climate and soil [32]. Path analysis reveals the components affecting BGBP, highlighting climate—specifically precipitation—as a key factor that influences soil nutrient dynamics and directly affects BGBP. Harsh, cold, and dry climates restrict nutrient availability, causing plants to allocate more biomass belowground for survival [33]. In contrast, warm and humid climates promote nutrient availability, favoring aboveground biomass expansion [34].

The regulatory mechanisms differ significantly between broadleaved and coniferous forests. In broadleaved forests, precipitation takes precedence, shaping biomass distribution both directly and through its influence on nutrient availability [33,35]. On the other hand, coniferous forests primarily achieve their biomass equilibrium through direct soil-related factors, with temperature playing a secondary but crucial role.

Broadleaved regions, characterized by angiosperms, employ unique evolutionary strategies that result in elevated productivity, setting them apart from the hardy conifers in alpine environments. The survival of conifers in these regions depends on soil nutrients and temperature regimes. Fertile areas with abundant precipitation provide a wealth of nutrients [5], directly and indirectly shaping biomass distribution in broadleaved forests. In contrast, coniferous landscapes prioritize nutrient allocation to their foliage, leading to increased photosynthetic efficiency. Notably, they allocate fewer nutrients to woody components, highlighting a survival strategy that emphasizes the enrichment of photosynthetic organs, which is crucial for thriving in nutrient-deficient environments.

## 4. Material and Methods

### 4.1. Experimental Sites and Sampling

To elucidate the spatial distribution of forest BGBP (belowground biomass proportion) and its driving factors, our investigation extended to 580 broadleaved and 345 coniferous forests across China from 2005 to 2020 (Figure 1A). To ensure methodological rigor, particularly considering the potential influence of mixed-species stands on biomass allocation, our analysis was focused exclusively on monospecific forests. Please refer to Appendix A for data sources, and note that our data assimilation procedures from the supplementary literature mirrored our experimental protocols. During fieldwork, we selected a minimum of four representative sample plots, each measuring 30 m × 30 m, to capture the typical zonal vegetation characteristics [5]. Geospatial coordinates, altitude, and slope information were meticulously recorded for comprehensive analysis. Our investigation primarily centered on tree species, as they are the primary contributors to forest biomass. In each quadrant, we documented specific tree details, including spatial positioning, diameter at breast height (DBH), and height, for all trees with a DBH exceeding 1 cm. BGBP was quantitatively determined by dissecting belowground biomass (BGB) relative to the total of aboveground biomass (AGB) and BGB.

### 4.2. Aboveground and Belowground Biomass Data

#### 4.2.1. Aboveground Biomass Data

In each plot, our study focused on carefully selected representative trees, with a cohort of 10 specimens per species (DBH > 5 cm). We dissected the aboveground tree structure into three main components: bark, foliage, and branches. The tree trunk was sectioned biennially, and individual segments were weighed to determine the cumulative fresh mass. Using the average standard branch technique, we identified typical branches, conducting successive weigh-ins to measure their fresh mass while simultaneously collecting and preserving the leaves in airtight containers with desiccants for accurate subsequent gravimetric analysis. After the fieldwork, all samples from the three components were transported to our laboratory, where they underwent a 48 h desiccation process at 70 °C [35]. This method allowed us to precisely determine the dry weight of tree trunks, branches, and foliage, facilitating the calculation of the total aboveground biomass (AGB) specific to each site.

#### 4.2.2. Belowground Biomass Data

We conducted a thorough excavation of each tree’s complete root system, reaching a soil depth ranging from 0 to 50 cm. Subsequently, roots and root clusters were isolated, with all attached soil carefully removed. The fresh mass of the roots was then determined. We selected specific root clusters weighing between 1000 and 2000 g and root segments within the 500–1000 g range for transport to our analytical facility. Despite the relatively minor contribution of fine roots (<5 mm) to biomass [36], and the inherent challenges in their comprehensive extraction, we persisted in our systematic efforts to collect a comprehensive array, diligently measuring the mass of each specimen obtained. To quantify belowground biomass (BGB), all collected root specimens underwent a desiccation process in a well-ventilated oven set at 85 °C until a consistent weight was achieved. This crucial step allowed us to calculate the dry mass, serving as the foundation for subsequent analytical extrapolations.

### 4.3. Plant Functional Traits and Environmental Factors

In our quest to decipher plant resilience strategies amid diverse ecological niches, we computed several quintessential leaf traits: leaf area (LA), specific leaf area (SLA), leaf nitrogen content (LN), leaf phosphorus content (LP), nitrogen-to-phosphorus ratio (N/P), and leaf dry matter content (LDMC) [37,38,39].

In each forested microcosm, a random assemblage of 20 or more mature, verdant trees was selected per species. Foliage specimens, unblemished and uniform, were delicately excised from the crowns and sandwiched between moistened filter papers, sequestered within Ziplock sanctuaries designated by species, and conserved under refrigeration.

LDMC was discerned by the quotient of foliar dry mass (in milligrams) and saturated fresh mass (in grams). Leaves, post refrigeration, were ensconced in darkness at 5 °C for a nocturnal phase to purge superficial hydration, followed by gentle blotting to absorb residual dampness. Precision weighing preceded and succeeded a 24 h desiccation sojourn at 80 °C.

Leaf scans, performed using a UNIS K3000C scanner, were subjected to Image-J software analysis to acquire LA metrics. SLA was expressed as the foliar surface (cm^2^) to desiccated mass (g) ratio. Subsequent elemental analyses involved pulverizing leaves for nitrogen and phosphorus assessments via the Kjeldahl and Mo-Sb colorimetric methodologies, respectively, detailed exhaustively in Pérez-Harguindeguy et al. [40].

Given the convolutions of theoretical predictions for individual plant traits, owing to both intra- and interspecies competitive dynamics [41], we embraced Community-1eans (CWM) to articulate average forest attribute values:(1)$$CWM=\sum_{i=1}^{S}D_{i}\times Trait_{i}$$

Herein, CWM epitomizes the averaged value of community-weighted traits, with “D_i_” denoting the prevalence of predominant arboreal taxa, and “Trait_i_” reflecting the specific traits under scrutiny [42].

Climatological constants, specifically mean annual temperature (MAT) and mean annual precipitation (MAP), were retrieved from WorldClim version 2.0 [38,43], boasting a 1 km^2^ resolution. Concurrently, soil metrics (pH, nitrogen, and phosphorus) within the inaugural 30 cm stratum were sourced from portals including https://www.osgeo.cn/data/wc137, which accessed on 31 October 2022.

### 4.4. Statistical Analysis

In our rigorous analytical approach, we harnessed the “agricolae” package within R to administer a *t*-test, anchoring the significance threshold at 0.05, thereby discerning disparities in belowground biomass production (BGBP) across broadleaved and coniferous forests. Concurrently, leaf functional traits underwent principal component analysis (PCA) through the “pcaMethods” package, with the inaugural two principal components (PC1 and PC2) distilled as trait proxies [44]. This strategic maneuver elucidated trait-BGBP correlations.

Our investigation transcended mere correlation, integrating abiotic parameters and inherent biological variables in explicating BGBP’s spatial oscillations, a feat actualized through a linear mixed-effects model, its congruence quantified via marginal R^2^ indices, using the “lme4” package [45,46]. The “vegan” package facilitated a Mantel test, crystallizing the nexus between constitutive/abiotic elements and BGBP, subsequently visualized as a heatmap [47].

A random forest methodology, leveraging the “randomForest” and “rfPermute” packages, was instrumental in identifying cardinal predictors of BGBP within divergent forest typologies, encompassing climatic variables, soil constituents, and salient leaf traits.

In disentangling the causal labyrinth governing BGBP spatial dynamics, we employed piecewise structural equation modeling (piecewiseSEM), scrutinizing each variable’s direct and ancillary impacts. Variables coalesced into composite groups, forming the bedrock of the structural equation modeling (SEM). This methodology, fortified against random sampling variances, yielded both “marginal” and “conditional” predictor insights [48,49] via R’s “piecewiseSEM,” “nlme,” and “lme4.” Model fidelity was corroborated through Fisher’s C test [50], with emphases on path coefficient significance (*p* < 0.05) and model robustness (0 ≤ Fisher’s C/df ≤ 2 and 0.05 ≤ *p* ≤ 1) [49], ensuring analytical refinement.

## 5. Conclusions

Our research illuminates a unique gradient in BGBP allocation within Chinese forests, declining from northwest to southeast. Remarkably, coniferous forests register a superior BGBP compared to their broadleaved counterparts. This biomass partitioning, dominant in both arboreal domains, is principally governed by climatic vectors, where precipitation asserts significant sway. Such climatic elements wield their influence predominantly through the recalibration of edaphic constituents, especially soil pH.

## Figures and Tables

**Figure 1 plants-12-03926-f001:**
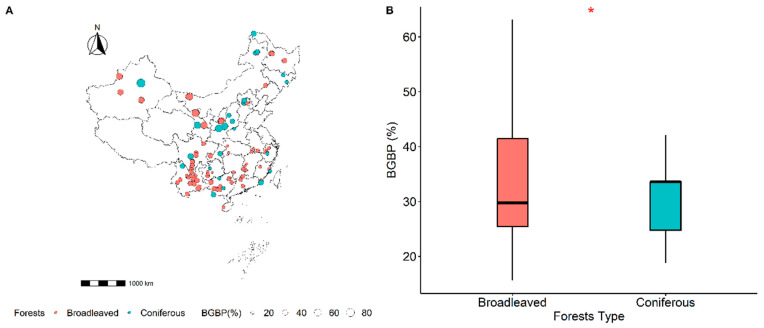
Spatial distribution of BGBP (%) and location of sampling sites in broadleaved and coniferous forests in China (**A**). Red circles represent broadleaved forests, and blue circles represent coniferous forests; the size of the circles indicates the relative value of BGBP (%). Comparison of BGBP (%) between broadleaved and coniferous forests, with significance assessed at the 0.05 level (**B**). * *p* < 0.05.

**Figure 2 plants-12-03926-f002:**
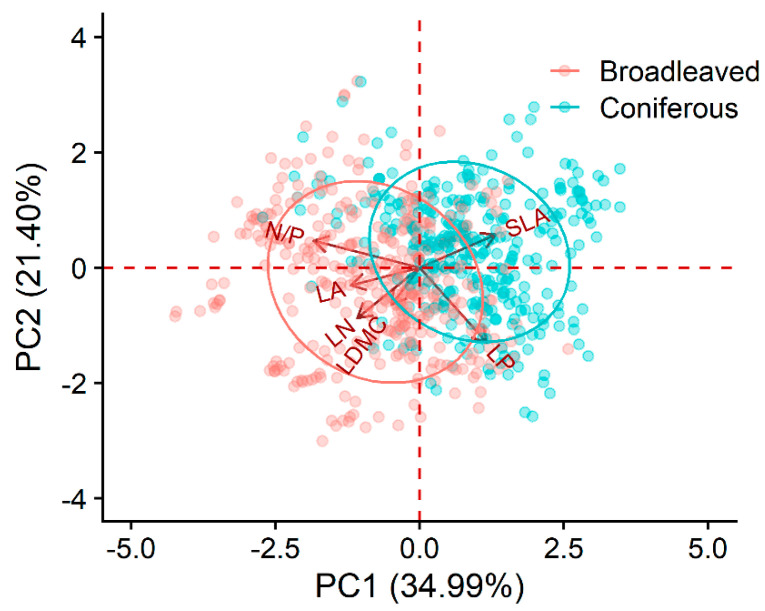
Principal component analysis (PCA) of functional traits in broadleaved and coniferous forests. Functional traits included leaf area (LA), specific leaf area (SLA), leaf dry matter content (LDMC), leaf nitrogen content (LN), leaf phosphorus content (LP), and leaf nitrogen to phosphorus ratio (N/P). All functional trait data were log-transformed. The first two principal components (PC1 and PC2) explained nearly 56.39% of the variation in functional traits.

**Figure 3 plants-12-03926-f003:**
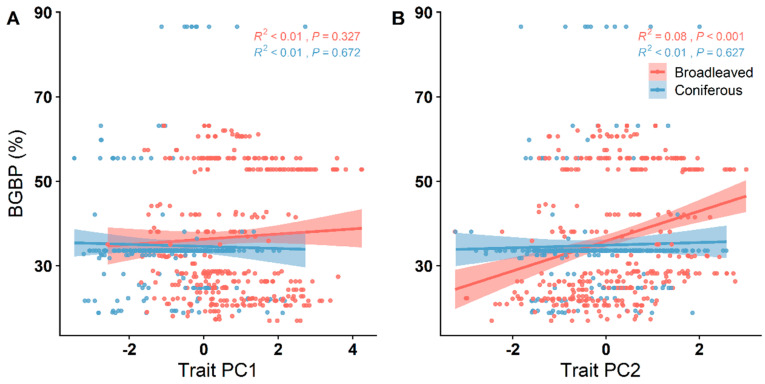
Linear relationships between BGBP (%) and the first two principal components of PCA of leaf functional traits. (**A**) The first principal component; (**B**) the second principal component. R^2^ represents the model goodness of fit, and *p*-values < 0.05 are statistically significant.

**Figure 4 plants-12-03926-f004:**
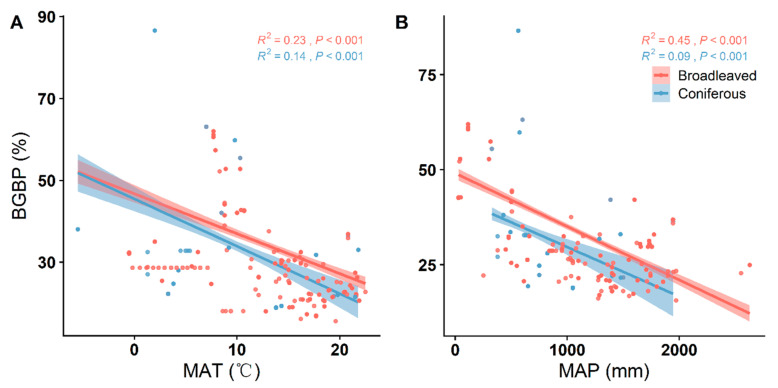
Linear relationships between climate factors and the BGBP (%) of broadleaved and coniferous forests. R^2^ represents the model goodness of fit, and *p*-values indicate significance. Climatic factors included (**A**) mean annual temperature (MAT); and (**B**) mean annual precipitation (MAP).

**Figure 5 plants-12-03926-f005:**
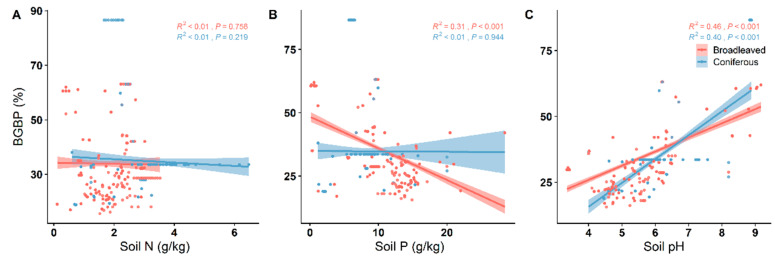
Linear relationships between soil nutrient factors and the BGBP (%) of broadleaved and coniferous forests. R^2^ indicates the model goodness of fit, and *p*-values indicate significance. Soil factors included (**A**) soil total nitrogen content (soil N); (**B**) soil available phosphorus content (soil P); and (**C**) soil pH.

**Figure 6 plants-12-03926-f006:**
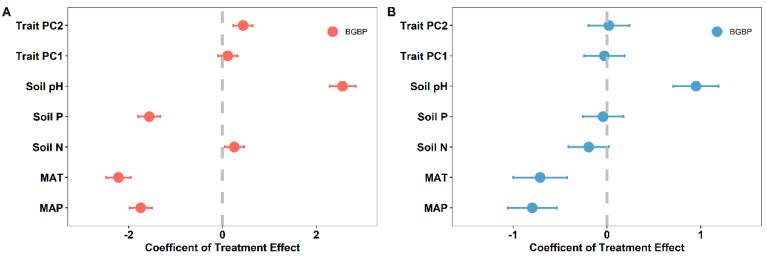
Explanatory effects of different treatments on BGBP (%) in broadleaved and coniferous forests. (**A**) Broadleaved forests; (**B**) coniferous forests. The effect coefficient for each explanatory variable was normalized to indicate the relative contribution of that variable to each regression model. The treatment effect coefficient indicates the positive or negative impacts of treatments on BGBP (%). Treatments included the first two principal components of community functional traits (trait PC1 and trait PC2), climate factors (MAT and MAP), and soil nutrient factors (soil N, soil P, and soil pH).

**Figure 7 plants-12-03926-f007:**
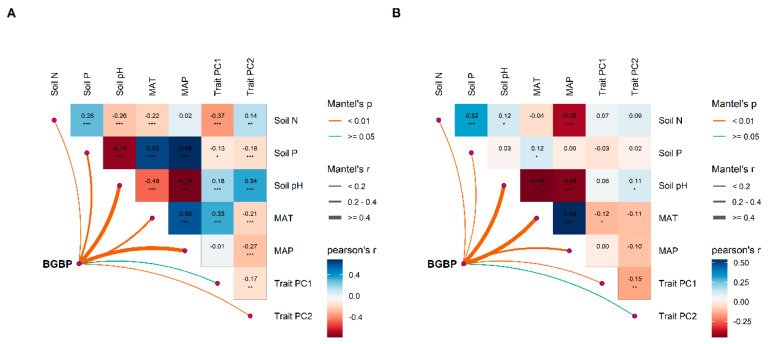
Multivariate correlation analysis of potential influencing factors of BGBP (%) in broadleaved forests (**A**) and coniferous forests (**B**). The influencing factors included the first two principal components of community functional traits (trait PC1 and trait PC2), climate factors (MAT and MAP), and soil nutrient factors (soil N, soil P, and soil pH). * *p* < 0.05, ** *p* < 0.01, *** *p* < 0.001.

**Figure 8 plants-12-03926-f008:**
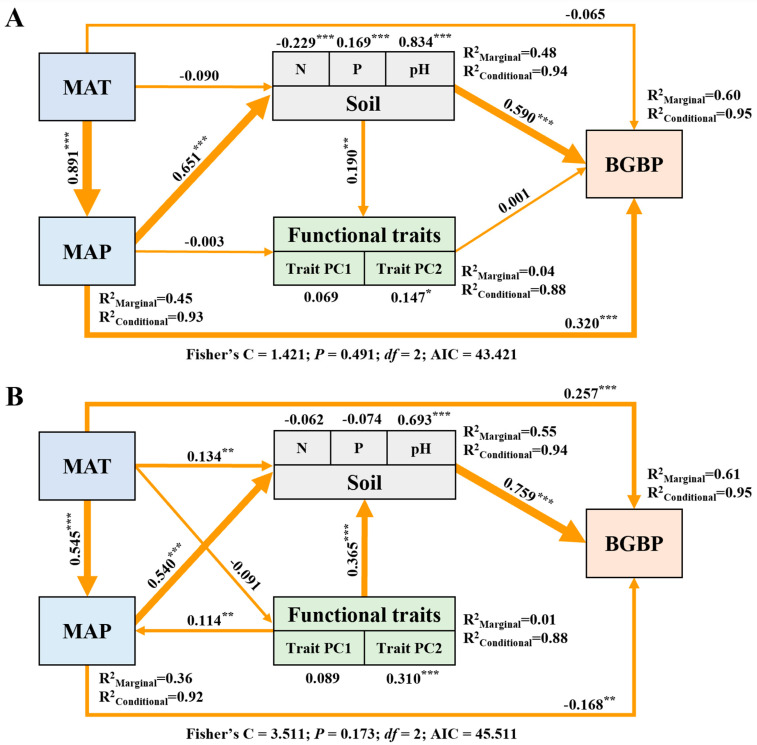
Linkages among climatic factors, soil nutrients factors, leaf functional traits, and BGBP (%) of broadleaved forests (**A**) and coniferous forests (**B**) in China. Path diagrams represent the standardized results of the final structural equation model (SEM) examining the relationships among variables. The number adjacent to the arrow is the path coefficient, which is the direct normalized effect size of the relationship. Asterisks indicate significance (*** *p* < 0.001; ** *p* < 0.01; * *p* < 0.05). R^2^ represents the generalized additive model (GAM) goodness of fit. The best SEM was selected with the lowest Akaike information criterion (AIC).

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
