# Peer review of "Climate Factors Affect Above–Belowground Biomass Allocation in Broad-Leaved and Coniferous Forests by Regulating Soil Nutrients"

_plants, 2023, doi:10.3390/plants12233926_

Round 1
Reviewer 1 Report (Previous Reviewer 3)
Comments and Suggestions for Authors
revised version is highly improved. Although some typo errors are still their such as scientifica formulas are not properly written and abbreviations are used multiple times etc. but these can be solved at Proof reading stage by authors.
Comments on the Quality of English LanguageOk
Author Response
Thank you very much for your valuable suggestions. We have corrected the errors.
Reviewer 2 Report (Previous Reviewer 2)
Comments and Suggestions for Authors
I have looked at the revision and it is improved. It is fine now to proceed.
Author Response
Thank you for the affirmation of our revision.
Reviewer 3 Report (New Reviewer)
Comments and Suggestions for Authors
The present manuscript, “Climate factors affect above-belowground biomass allocation in broad-leaved and coniferous forests by regulating soil nutrient” by Zhang et al. , studies the differences in belowground biomass allocation of broadleaved and coniferous forests along a geographical gradient in China. Authors have investigated a very acceptable number of study sites over a period of 15 years, which entails a knowledge of value for the scientific community. Since climatic factors and soil nutrients influence biomass allocation, authors provide with a good analysis on the effect of annual mean temperature and precipitation, as well as of soil P, N and pH on BGBP; they also relate in their study these environmental factors with plant functional traits. In this regard, the present study is of scientific interest. On the other hand, authors fail in providing results of how these environmental factors also influence the aboveground biomass, or it is not clear along their manuscript (i.e., there is a 2.2.1 subsection that explains how the aboveground biomass data was obtained but there is not any reference to the ABG in the results section). Therefore, this part of the study, which the tittle also references to, is missing. In other words, either the tittle should be change and this part of the material and methods avoided, or better, to supply with data this part of the study, and discuss/integrate it in the manuscript.
Another shortcoming is the type of writing along the manuscript. The present manuscript shows an enormous excess of wordy and pompous writing, which is not suitable with scientific writing. The rich and grandiloquent vocabulary used in the manuscript would fit probably beautifully in the context of literary writing, but not in a scientific context. Scientific writing should be straightforward and concise, and not ornated up to this magnitude. I am attaching the pdf file highlighting some (but only some) of the expressions/words that are not fitting in this scientific context. All this to say that, in my opinion, much of this unusual style for a scientific article should be re-edit.
Additionally, below are a few specific comments that authors should clarify or deal with to meet scientific rigor:
L3: soil nutrients
L38-50: The R/S ratio has not been mentioned along the manuscript after this introductory paragraph (i.e., in the results or discussion). In my opinion, if this is not relevant to justify the interest of the present study, authors should avoid this paragraph. Introduction should focus on justifying the importance of the objectives considered in the study and should be aligned with the thread of its content.
L 42: Define pure coniferous and broadleaved forests to distinguish from those subsequent mentioned: natural or planted.
L 44: Deciduous coniferous forests, is this correct? Do you have deciduous coniferous forest in the biogeographical area studied here?
L151: Do you mean “within the evolving global change panorama”?
L155-168: too many pompous words for a Material and Methods section and a scientific document: epitomizing, crystallized, heft?
L160-161: Clarify how many plots. Material and Methods section must be potentially reproducible to allow comparisons among the scientific community, and therefore, should be very rigorous explained.
L164: what do you mean with quadrant? Plot? Decide how to name the plots of 30m x 30m; or define quadrant, and how many were in each plot.
L166: “computations of species diversity”. This is very confusing because the authors inquiry was related with trees, and they indicate that the forests were monospecific.
L172: One citation is missing here? Where this information shown in the map comes from? If this is author´s information from their investigation for the present study, then it should be placed in the results section.
L285-288: Fig. 3. Please check if values for P and R2 are properly assigned as they are shown. To me, they seem to be the other way around because values of R2 range between 0-1. The same for Fig. 5A.
L317: pH and MAT perhaps?
L318: MAT and MAP both, right?
L347: shouldn´t be also a subsection of Aboveground Biomass in the results? See 2.2.1 Aboveground biomass data.
INTRODUCTION the highest shortcoming of this section is that in many parts is missing connection with the results presented, and citations are few. It needs more work. Authors should connect some of the paragraphs referred to other studies with their results and vice versa:
L 365: Add “in our study” after (MAP) or somewhere in this paragraph. Authors´s results corroborate those previous finding and should say so.
L368: is redundant and wordy, it should be erased as it is indicated in the attached pdf.
L373: “realms” better substituted by woodlands as it is more suitable in this context.
L374: Clarify what type of biomass productivity. Aerial?
L374-383: The content shows a contradictory argument if looking at Fig. 1B. BGBP is higher (although more variable) in broadleaved forests. Please rewrite and clarify this information and add some citations that would corroborate (or not) your findings.
L379: “floras” is not specific in this context, better to refer to “trees”?
L 383-384: Broadleaved is not a phylum botanically, and neither a relative to conifers. Delete phylum, and substitute relatives by woodlands.
L385-403: Are there other studies that relate plant functional traits differences between broadleaved and conifers woodlands to discuss your results? Citations?
L403.404: Neither “tremors” or “brethren” are appropriate or scientific terms here.
L405-418: Is this paragraph relevant to discuss your results? If not, it should be erased. Such information needs connection with your results (for example, it is well done in the following paragraph L420-429). At the present form the content looks like a copy and paste with no discussion in it.
L430-444: this information should include some mentions to your results. Again, this needs to be connected or following a thread with your findings.
L448: “bows” and “diktats” are not common terms in scientific writing.
L 461-462: this finding is relevant and should be discussed above in the discussion.

Comments on the Quality of English Language
The present manuscript shows an enormous excess of wordy and pompous writing, which is not suitable with scientific writing. Scientific writing should be straightforward and concise, and not ornated up to this magnitude. I am attaching the pdf file highlighting some of the expressions/words that are not fitting in this scientific context, and I do recommend to re-edit the manuscript.
Author Response
Reviewer #3 (Remarks to the Author):
The present manuscript, “Climate factors affect above-belowground biomass allocation in broad-leaved and coniferous forests by regulating soil nutrient” by Zhang et al. , studies the differences in belowground biomass allocation of broadleaved and coniferous forests along a geographical gradient in China. Authors have investigated a very acceptable number of study sites over a period of 15 years, which entails a knowledge of value for the scientific community. Since climatic factors and soil nutrients influence biomass allocation, authors provide with a good analysis on the effect of annual mean temperature and precipitation, as well as of soil P, N and pH on BGBP; they also relate in their study these environmental factors with plant functional traits. In this regard, the present study is of scientific interest. On the other hand, authors fail in providing results of how these environmental factors also influence the aboveground biomass, or it is not clear along their manuscript (i.e., there is a 2.2.1 subsection that explains how the aboveground biomass data was obtained but there is not any reference to the ABG in the results section). Therefore, this part of the study, which the tittle also references to, is missing. In other words, either the tittle should be change and this part of the material and methods avoided, or better, to supply with data this part of the study, and discuss/integrate it in the manuscript.
Another shortcoming is the type of writing along the manuscript. The present manuscript shows an enormous excess of wordy and pompous writing, which is not suitable with scientific writing. The rich and grandiloquent vocabulary used in the manuscript would fit probably beautifully in the context of literary writing, but not in a scientific context. Scientific writing should be straightforward and concise, and not ornated up to this magnitude. I am attaching the pdf file highlighting some (but only some) of the expressions/words that are not fitting in this scientific context. All this to say that, in my opinion, much of this unusual style for a scientific article should be re-edit.
Additionally, below are a few specific comments that authors should clarify or deal with to meet scientific rigor:
Response: Thank you very much for your feedback. Since our research focuses on the allocation patterns of biomass above and belowground, BGBP represents the proportion of belowground biomass, and the sum of aboveground and belowground proportions equals 1. Therefore, further discussion on AGBP may not be meaningful. Additionally, we have made modifications and refinements to the language part as per your suggestions. We greatly appreciate your feedback.
L3: soil nutrients
Response: We have corrected it.
L38-50: The R/S ratio has not been mentioned along the manuscript after this introductory paragraph (i.e., in the results or discussion). In my opinion, if this is not relevant to justify the interest of the present study, authors should avoid this paragraph. Introduction should focus on justifying the importance of the objectives considered in the study and should be aligned with the thread of its content.
Response: We have corrected it. In this study, the meanings represented by BGBP and R/S are essentially the same. Because BGBP is equal to belowground biomass divided by total biomass, while R/S is equal to belowground biomass divided by aboveground biomass.
L42: Define pure coniferous and broadleaved forests to distinguish from those subsequent mentioned: natural or planted.
Response: We have defined them:forest composed solely of coniferous species or broadleaf species.
L44: Deciduous coniferous forests, is this correct? Do you have deciduous coniferous forest in the biogeographical area studied here?
Response: Thanks for your suggestions. It should be coniferous forests
L155-168: too many pompous words for a Material and Methods section and a scientific document: epitomizing, crystallized, heft?
Response: Thanks for your suggestions. We have revised them.
L160-161: Clarify how many plots. Material and Methods section must be potentially reproducible to allow comparisons among the scientific community, and therefore, should be very rigorous explained. what do you mean with quadrant? Plot? Decide how to name the plots of 30m x 30m; or define quadrant, and how many were in each plot.
Response: Thanks for your suggestions. Due to the large geographic scale we selected, the number of plots varied for each location, so we provided a range.
L166: “computations of species diversity”. This is very confusing because the authors inquiry was related with trees, and they indicate that the forests were monospecific.
Response: Thanks for your suggestions.We have excel the line 166.
L285-288: Fig. 3. Please check if values for P and R2 are properly assigned as they are shown. To me, they seem to be the other way around because values of R2 range between 0-1. The same for Fig. 5A.
Response: Thanks for your suggestions. After the check, the figures are right.
L347: shouldn´t be also a subsection of Aboveground Biomass in the results? See 2.2.1 Aboveground biomass data.
Response: We calculated the proportion of belowground biomass; therefore, discussing the proportion of aboveground biomass and belowground biomass is essentially the same.
L365: Add “in our study” after (MAP) or somewhere in this paragraph. Authors´s results corroborate those previous finding and should say so.
Response: We have added in our study.
L368: is redundant and wordy, it should be erased as it is indicated in the attached pdf.
Response: We have excel this sentence.
L373: “realms” better substituted by woodlands as it is more suitable in this context.
Response: Thanks for your advice. We used realms.
L379: “floras” is not specific in this context, better to refer to “trees”?
Response: Thanks for your advice.
L403.404: Neither “tremors” or “brethren” are appropriate or scientific terms here.
Response: We have corrected them.
L405-418: Is this paragraph relevant to discuss your results? If not, it should be erased. Such information needs connection with your results (for example, it is well done in the following paragraph L420-429). At the present form the content looks like a copy and paste with no discussion in it.
Response: We have enhanced the discussion in this regard.
L448: “bows” and “diktats” are not common terms in scientific writing.
Response: We have corrected them.
L 461-462: this finding is relevant and should be discussed above in the discussion.
Response: We have added the discussion of this part.
**************************** END **************************
This manuscript is a resubmission of an earlier submission. The following is a list of the peer review reports and author responses from that submission.
Round 1
Reviewer 1 Report
Comments and Suggestions for Authors
The paper reports the response of coniferous and broadleaved forest to climate factors, analyzing biomass allocation.
This is a relevant area of research and is within the scope of Plants; however, I have a hard time trying to understand the novelty of the study, since more or less the same results (as stated by the authors) could be found in the literature.
The authors talk about biotic factor (line 152 and in other places), but I think is more appropriate species-specific factors or constitutive factors. Biotic factor is used for the effect of another organism.
Methodology:
Is there a reason to limit the root biomass to 50 cm depth? Do the root depth distribution change between species?
Statistical analysis:
More info should be showed.
The R functions were performed with everything in default or some changes were made?
Line 284-285: Which test did you use?
Fig 2. A PCA should not be performed with completely correlated variables because these variables are overrepresented. In this case, for example, the SLA and the LDMC, as one is the inverse of the other.
Minor comments.
Line 147: Remove conversely, as it does not counter an idea.
Fig 1 A: Try to improve the quality of the figure
Fig 1 B: Please, explain how the figure is made. The meaning of the horizontal bar, if it is the median or the mean, and the meaning of the vertical bars. It could be interesting to add the data as dots to better visualize the data.
Line 277: Di and Traiti, the i as subindex
Figs 4 and 5. Use the same scale in the y-axes
Line 456-457: It is repeated.
Reviewer 2 Report
Comments and Suggestions for Authors
The manuscript overall is good and contributes to our knowledge.
---My biggest suggestion for improvement is that the writing is rather inefficient and the whole manuscript could be shortened without much loss of information. This applies to all the sections, especially the Intro and Discussion sections.
-For example, lines 47-53 now read:
“On a global scale, the allocation patterns of plant biomass reflect the intricate strategies employed by vegetation to acquire and manage above-ground and belowground resources (Ma et al., 2021). This allocation strategy is a pivotal adaptation mechanism for terrestrial ecosystems, enhancing their carbon sequestration capabilities (Qi et al., 2019). The study of plant biomass allocation holds substantial significance in the context of comprehending and predicting carbon storage within terrestrial ecosystems (Zhou et al., 2022).”
-This could be rewritten to:
“Biomass allocation globally reflects strategies to acquire and manage above- and belowground resources (Ma et al., 2021), a crucial adaptation for enhancing terrestrial carbon sequestration capabilities (Qi et al., 2019). Understanding and predicting terrestrial carbon storage can be improved through studying plant biomass allocation (Zhou et al., 2022).”
-Similarly in the Discussion: lines 428-432 now read:
“Broadleaved forests predominantly consist of angiosperm broadleaved tree species, setting them apart from coniferous forests composed of coniferous tree species (Gao et al., 2023). These two forest types exhibit significant disparities in...... environmental niches, as highlighted by Li et al. (2021).”
-This could be rewritten to:
“Broadleaved angiosperm forests and coniferous forests exhibit significant disparities in..... environmental niches (Li et al., 2021, Gao et al., 2023).”
I don’t think specifying that coniferous forests are composed of coniferous trees is worth specific mention, for example. There is a lot of this unnecessarily bulky writing.
---Other types of needless or odd verbiage include (these are just some examples):
-line 15, delete the word spanning. The sentence is fine without it.
-line 45, comprehending is a strange use of this word. Perhaps understanding?
-line 97, please change the word milder to something better.
-lines 217, 230, 231, 237 (and elsewhere), the word meticulously is used repeatedly. Carefully is enough here. Meticulous is an extreme version of careful. A brain surgeon does meticulous work. An ecologists with dirt does careful work. When dealing with dirt and roots, and digging holes in the ground, meticulousness seems improbable.
-lines 438 (On one hand) and 442 (On the other hand) are colloquial and should be rephrased.
OTHER COMMENTS
---Line 24, define and these functional traits. Line 26, same thing for climatic factors. Identify them here briefly.
---Lines 38-40, need a reference
---Line 41, is it living organisms or living plants? You don’t include a bird and its chicks?
---Lines 47-48, I’m not sure I understand the concept of ‘intricate’ here when discussing things at a global scale.
---Line 108, please define optimal allocation theory. If it is the verbiage that follows, then write, “...theory, which states that....”
---Line 192 vicinity, perhaps a short section on study site. This would be a good place to describe the vegetation gradient across China briefly including dominant species, community types, etc. Nothing too long or extensive, but some background information on the systems being studied.
---Lines 201-202: how were the plots “randomly chosen?” How was randomness established?
---Fig 1B, I’m not sure why there is an asterisk in the top center.
---Line 412, first explain the main results and connection to the literature. Then get into the implications such as climate change. Please talk about the findings first, then how they fit into the bigger picture of climate change.
---In addition to the above comments about the writing, I found the density of the writing to also make it difficult to extract main points. Lots of ‘this is important’ and ‘that is important,’ but more complete thoughts would be even better. Even the Conclusion is so full of generalities that it’s hard to extract the big messages for the reader to take away. Complete sentences bearing all of the information, variables specifically being identified, etc. should be here! For example, line 538-540, what specifically about above and belowground biomass (list the variables), and which central role (doing what specifically) and relating in which ways (the nature of the relationship) to which specific climate factors (temp? precip?)?
Comments on the Quality of English Language
I have addressed this in my comments above.
Reviewer 3 Report
Comments and Suggestions for Authors
The article “ Climate factors affect above-belowground biomass allocation in broad-leaved and coniferous forests by regulating soil nutrient “ is well written, provide good results. But presentation is bad and writing needs to be improved.
For example
Keywords, replace all these, use symmetrically and donot use long keywords, and these should not be included in the title.
Introduction. It is too long and too boring reduce is to half at least. Line 72-175 so much information repeating and un necessary. This is a research article not a prospective …so avoid this and reduce this.
Results and methods are good and well explained
Line 430-460, poor discussion, improve this part
Conclusion. It is short, suggested to remove this part.
Comments on the Quality of English Language
ok
Reviewer 4 Report
Comments and Suggestions for Authors
The manuscript is written very well. However, I have some concerns regarding the methods and the conclusions drawn.
Line 25-27: Need to be specific regarding the climatic factors.
Line 198: Need to explain what a pure forest is.
Section 2.2.1: I am unsure of the sampling method, it would be nice if the authors provided some references. It would also help the authors provide some information on the age of the trees, average height, and average DBH. What was the range of DBH? Just stating DBH>5cm is too vague. If you are drying above-ground vegetation including tree trunks is 70oC for 48 hours sufficient?
Section 2.2.2: Need to add more information on the root sample collection method. What was the excavated diameter?
Lines 238-239: Why is there a difference in the drying of above-ground and below-ground parts? Shoots you are drying at 70°C for 48 hours and roots 85°C until they reached a consistent weight, why not use the same method?
Section 2.3: Why not use the leaves from the plants selected for collecting biomass?
LInes 259: 260: Authors need to clarify why different drying techniques are used.
Some additional questions:
Why was Soil moisture, a key factor influencing plant biomass allocation to roots, not included in the study?
There is high spatial variability in soil nutrient status. Why not analyze the soil excavated during root sample collection?